# Effects of Doxycycline Treatment on Hematological Parameters, Viscosity, and Cytokines in Canine Monocytic Ehrlichiosis

**DOI:** 10.3390/biology12081137

**Published:** 2023-08-16

**Authors:** Saulo Pereira Cardoso, Adenilda Cristina Honorio-França, Danielle Cristina Honorio França, Luana Paula Sales Silva, Danny Laura Gomes Fagundes-Triches, Maria Clara Bianchini Neves, Aron Carlos de Melo Cotrim, Arleana do Bom Parto Ferreira de Almeida, Eduardo Luzía França, Valéria Régia Franco Sousa

**Affiliations:** 1Instituto Federal de Educação, Ciência e Tecnologia de Mato Grosso, Barra do Garças 78607-899, MT, Brazil; saulo.cardoso@ifmt.edu.br; 2Programa de Pós-Graduação em Ciências Veterinárias, Faculdade de Medicina Veterinária (FAVET), Universidade Federal de Mato Grosso (UFMT), Cuiabá 78060-900, MT, Brazil; mariaclarabianchini1@gmail.com (M.C.B.N.);; 3Programa de Pós-Graduação em Imunologia e Parasitologia Básicas e Aplicadas (PPGIP), Universidade Federal de Mato Grosso (UFMT), Barra do Garças 78605-091, MT, Brazilluanasalesvet@hotmail.com (L.P.S.S.); dannylauragf@hotmail.com (D.L.G.F.-T.); aroncarlosbg@gmail.com (A.C.d.M.C.); eduardo.franca@ufmt.br (E.L.F.)

**Keywords:** *Ehrlichia canis*, leukocytes, platelets, dogs, doxycycline

## Abstract

**Simple Summary:**

Canine Monocytic Ehrlichiosis is a disease of dogs caused by intracellular bacteria transmitted through tick bites. This disease can cause changes in a dog’s blood, affecting the count of cells and components of the defense system, such as cytokines, as well as the viscosity of the blood. This study was conducted to explore these changes in naturally infected dogs with ehrlichiosis, who were untreated and treated with a doxycycline-based antibiotic at a dose of 10 mg/kg every 12 h for 28 days. Even after treatment, infected dogs exhibited a decrease in blood viscosity. The infection decreased total white blood cells, lymphocytes, and the cytokine tumor necrosis factor-alpha level and increased cytokine interleukin-1-beta. Infected dogs showed a correlation between cytokines interleukin 10 and 12 with blood viscosity. Treating dogs with monocytic ehrlichiosis with doxycycline can help restore blood parameters such as platelets and eosinophils, but it may also elevate levels of interleukin-1-beta and monocytes. Therefore, assessing viscosity and cytokine levels is vital when treating dogs with this condition.

**Abstract:**

This study aimed to analyze the hematological parameters, blood viscosity, and cytokines of dogs infected by *Ehrlichia canis* untreated and treated with doxycycline. Initially, 47 dogs were examined, and 36 were suspected to have canine monocytic ehrlichiosis, which was confirmed through molecular polymerase chain reaction tests. This study consisted of 25 dogs, with 11 being healthy and 14 testing positive for *E. canis*. The dogs were divided into experimental groups based on their test results, including a control group of healthy dogs (*N* = 11), a group of infected dogs without treatment (*N* = 7), and a group of infected dogs treated with doxycycline (*N* = 7) at a 10 mg/kg dose every 12 h for 28 days. Blood samples were taken to determine hematological parameters, viscosity, and cytokine levels. It was observed that, regardless of doxycycline treatment, there was a reduction in total leukocytes and lymphocytes in infected dogs with *Ehrlichia canis.* The eosinophils and platelets decreased in dogs with *Ehrlichia canis* infections without treatment. Monocytes, eosinophils, and platelets increased when the dogs were treated with doxycycline. Regardless of treatment, infected dogs’ blood viscosity was lower than uninfected dogs. Infected dogs showed lower TNF-α and increased IL-1β. There was a correlation between the blood viscosity with the cytokines IL-10 and IL-12 in the infected dogs. The eosinophil count correlated with TNF-α in the group of infected and untreated dogs. In conclusion, treating dogs with monocytic ehrlichiosis using doxycycline can increase platelet and eosinophil levels but may also increase IL-1β and monocyte levels, exacerbating inflammation. Therefore, evaluating viscosity and cytokine levels is important when treating dogs with this condition.

## 1. Introduction

Bacteria of the Anasplasmataceae family can infect blood cells such as erythrocytes, leukocytes, or platelets, leading to systemic disorders in animals and humans [1]. One such disorder is Canine Monocytic Ehrlichiosis (CME), caused by an obligate intracellular Gram-negative bacterium, *Ehrlichia canis* (*E. canis*), which has an affinity for monocytes. This bacterium is commonly found in domestic dogs and is transmitted through bites from hard ticks (Ixodidae) such as *Rhipicephalus sanguineus*. CME is a globally distributed disease in tropical and subtropical regions [2], with Brazil being one of the countries where the disease is prevalent. Studies have shown that CME is widely disseminated in Brazil [3,4], and the rates of *E. canis* infections in dogs vary across different regions of the country, resulting in a prevalence of the disease between 22% and 76% [5,6,7].

CME evolves with different clinical signs. In the acute phase, the following can be observed: fever, serous or purulent oculonasal discharge, anorexia, weight loss, dyspnea, lymphadenopathy, splenomegaly, pale mucous membranes, hemorrhages, neurological signs, thrombocytopenia, edema, uremia, jaundice, and lameness [8]. In the subclinical phase, clinical signs may often not be observed or not detected during the clinical evaluation due to the apparent physical health of the dog [8]. Still, splenomegaly and intermittent fever may occur, or even signs not commonly reported by the dog’s owners [9]. Finally, the chronic phase can be mild or severe and is characterized by clinical signs of bleeding tendencies [10], depression, weight loss, lymphadenopathy, thrombocytopenia, anemia, pancytopenia [8], polyarthritis, and severe ocular injuries [9].

To cause infection, *E. canis* needs to escape elimination by the immune system, causing disturbances in the cellular environment and modulating the production of cytokines to favor persistent infection. The survival of the bacteria in the host depends on the inhibition of the immune response of the T helper 1 (Th1) type. The cytokine interferon-gamma (IFN-y), linked to the Th1 response, activates infected monocytes to eliminate intracellular microorganisms [11].

*E. canis* infection can lead to significant hematological abnormalities in dogs, including pancytopenia [12,13]. In addition, studies have shown that individuals with anemia [14], altered blood pressure [15], hyperviscosity syndrome [16], and diabetes [17] experience changes in blood viscosity. Similarly, dogs infected with bacteria from the Anaplasmataceae family demonstrated decreased viscosity [18]. However, the effects of *E. canis* infection on blood viscosity in animals have yet to be fully understood.

Blood viscosity is crucial in maintaining systemic arterial pressure and gasometry [19], as it helps maintain the blood’s rheological properties and restore homeostatic conditions [20]. In addition, infections, such as malaria, can cause changes in blood viscosity, and the interaction with cytokines may act as an immunomodulatory agent during the infection [21]. Dogs infected with *E. canis* showed changes in cytokine levels, which may be related to hematological changes in the disease.

Also, studies indicate that doxycycline is a highly effective drug for treating *E. canis* infections [22,23,24,25] because of its capacity to penetrate cells. Moreover, no research establishes a connection between the effects of doxycycline on viscosity and cytokines. Therefore, this study aimed to analyze the hematological parameters, blood viscosity, and serum cytokines of dogs infected by *E. canis* and either untreated or treated with doxycycline.

## 2. Materials and Methods

### 2.1. Animals

Blood samples were collected from 47 dogs, with different ages (>1 year) and no distinction of sex or breed that were haphazardly selected and evaluated in the city of Barra do Garças, State of Mato Grosso, Brazil (−15.8891, 15°53′24″ South, −52.2634, 52°15′24″ West). Regardless of disease stage, dogs with at least three clinical signs of CME were initially subjected to rapid serological testing. Of the total, 36 dogs with suspected CME were positive in commercial serological test kits (Alere^®^, Waltham, MA, USA). Next, *E. canis*-positive animals were tested via PCR to detect *E. canis*-specific DNA. All animals positive in rapid serological tests and confirmed by PCR were included in the study. Dogs that showed clinical signs of CME but tested negative in the PCR test, dogs that tested positive for *Leishmania* sp., *Babesia* sp., dogs under one year old, and dogs whose owners did not give permission were not included in the study. Finally, dogs were treated for CME with doxycycline (Syntec, São Paulo, Brazil) at 10 mg/kg every 12 h for 28 consecutive days [25]. Seven dogs returned for blood collection immediately after completing doxycycline treatment. According to the results of the diagnostic tests, clinical signs of the disease, and treatment adherence, the following groups were formed: dogs in the control group, negative for *E. canis* in both tests and without clinical EMC signs (*N* = 11); dogs infected by *E. canis,* reactive in both tests, and untreated (*N* = 7); and dogs reactive in both tests for *E. canis* and treated with doxycycline (*N* = 7). The scheme for obtaining samples and experimental design is described in Figure 1.

### 2.2. Serological Tests

Serological tests were applied using whole blood with EDTA. In addition, the Ehrlichiosis Ac Test Kit–Alere test^®^ (Waltham, MA, USA), which has immunochromatographic technology, was applied in dogs. The tests were used following the guidelines of their respective manufacturers.

### 2.3. Polymerase Chain Reaction (PCR)

DNA extraction from whole blood was performed using the phenol-chloroform method [26]. The DNA was dissolved in 20 mg/mL of ultrapure water and stored at –20 °C. To ensure the DNA’s purity and lack of inhibitors, we tested the samples using a PCR method to detect the canine b-globin gene. This gene amplifies a 119 bp fragment [27]. The DNA detection process used primers E_can0503F (5′-CAG CAA ATT CCA ATC TGC ACT TC-3′) and E_can0503R (5′-GAG CTT CCA ATT GAT GGGTCT G-3′) in which the Ecaj_0503 gene encodes 146 base pairs of a hypothetical protein (E_can0503 system) [28]. Qualitative PCR analysis was performed using a StepOne™ Real-Time PCR System Sequence Detection System (Thermo Fisher Scientific, Waltham, MA, USA). Reactions were prepared in a final volume of 25 µL containing non-specific double-stranded DNA intercalators (SYBR Green^®^ Master Mix, Thermo Fisher Scientific, Waltham, MA, USA), 0.3 µM of each primer, and 2 µL of target DNA. The reaction program consisted of the following steps: 94 °C for 10 min, followed by 40 cycles of 94 °C for 15 s for denaturation, and 60 °C for 30 s for annealing and extension.

Positive control was an animal positive for *E. canis* in PCR with morulae in blood smear, and negative control (DNA-free reaction) was included in all PCR. Amplified products were subjected to 1.5% agarose gel electrophoresis. They were all stained with Gel Red and visualized in a transilluminator (UV-300 nm) Chemi-doc (Bio-Rad TM, Berkeley, CA, USA).

### 2.4. Hematological Parameters

Whole blood samples with EDTA from the dogs were sent for laboratory analysis to obtain hematological data such as erythrogram, leukogram, and platelet count in an automated device Icounter Vet, model D Check D Plus (DIAGNO, Belo Horizonte, Brazil), according to the literature [29]. In addition, reference values were considered, as proposed in the literature [30].

### 2.5. Hemorheological Parameters

Whole blood with EDTA from each dog was subjected to hemorheology analysis to obtain data on blood viscosity. A compact modular rheometer, Anton-Paar^®^ Cone-Plate model–MCR 102 (Anton Paar^®^ GmbH, Ostfildern, Germany), was used, and the graphs were obtained using the Rheoplus software. For hemorheology analysis, 750 μL of whole blood with EDTA was used, maintaining a temperature of 37 °C in the plate with 60 flow measurement points [17].

### 2.6. Cytokines 

The dog’s blood serum samples were submitted for preparation for flow cytometry examination. Pro-inflammatory and regulatory cytokines were measured, such as IL-1β, IL-6, IL-8, IL-10, IL-12, and TNF-α [14,21]. This process was carried out with a Flow Cytometer, model FACScalibur (BD Biosciences, St. Louis, MI, USA), with specific marking using antibodies for each cytokine. Cytometric Bead Array (CBA) kits (BD Biosciences, St. Louis, MI, USA) were used for this quantification, and the procedure was executed as per the instructions provided by the manufacturer.

### 2.7. Statistical Analysis

Data were expressed as a mean and standard error. We analyzed the differences in blood properties and cytokine levels between noninfected, infected, and infected animals treated with doxycycline. A D’Agostino normality test and variance analysis (ANOVA) were used, followed by Tukey’s test between animals noninfected, infected, and infected and treated. The Student t-test for two related samples was used to compare before and after dogs treated with doxycycline. Additionally, we analyzed correlations between blood properties, cytokine levels, and hematological parameters using Pearson’s linear correlation. Biostat 5.0 was the software used for statistical analysis. If the *p* value is below 0.05, it indicates statistical significance.

## 3. Results

Out of the 36 dogs with clinical signs of CME, 9/36 displayed pyrexia, 9/36 showed apathy, 5/36 had pale mucosa, 5/36 suffered from hyporexia, and 4/36 had hemorrhages and lymphadenopathy. Furthermore, 2/36 dogs experienced vomiting, while 2/36 showed weight loss and diarrhea. Additionally, 1/36 of the dogs had onychogryphosis, stomatitis, and seizures. 

Regardless of doxycycline treatment, there was a reduction in total leukocytes and lymphocytes in infected dogs (Table 1). In addition, the eosinophils and platelet concentrations decreased in dogs with *E. canis* infections without treatment. Conversely, there was an increase in monocytes, eosinophils, and platelets in dogs infected after the treatment of doxycycline.

There was no significant difference in the erythrocytes, hemoglobin, hematocrit, leukocytes, neutrophils, and lymphocyte levels between dogs infected with and without treatment (*p* > 0.05). However, the doxycycline treatment increased the monocytes, platelets, and eosinophils (Figure 2) and restored the levels to similar values found in noninfected dogs (Table 1).

Figure 3 shows the blood viscosity of dogs non-infected or infected with *E. canis* treated or not with doxycycline. The mean blood viscosity of animals infected by *E. canis* decreased independently of treatment (Figure 3). The blood viscosity in dogs infected with untreated *E. canis* and treated with doxycycline is shown in Figure 4. The treatment with doxycycline reduced the blood viscosity in dogs with infection.

Table 2 shows the cytokine levels in dogs infected with *E. canis* serum. The IL1-β level was higher in dogs with CME, and there was no difference between treated and untreated dogs. The TNF-α, regardless of the treatment, showed lower concentrations in the serum of infected dogs. The lowest levels of TNF-α were in infected and untreated dogs. There was no significant difference in the concentrations of IL-6, IL-8, IL-10, and IL-12 between the groups studied.

After treatment with doxycycline, an increase in IL-6 concentrations was observed in infected dogs (Figure 5). However, no difference was observed without and with treatment for the other cytokines evaluated.

The cytokines were correlated with hematological parameters. IL-12 showed an inversely proportional correlation with hematocrit in infected and untreated dogs. There was a directly proportional correlation between TNF-α and IL-12 and the eosinophils in the untread group. In the infected and treated dogs group, the leukocyte parameter showed a strong and inversely proportional correlation with cytokine IL-10 (Table 3).

The group of uninfected dogs showed an inversely proportional correlation between TNF-α and blood viscosity. The IL-10 correlated inversely with blood viscosity in the group of infected and untreated dogs and correlated directly proportionally in the dogs treated with doxycycline. Cytokines IL-1β and IL-12 showed an inversely proportional correlation with blood viscosity in infected dogs not treated with doxycycline (Table 4).

Pearson’s linear correlations between the hematological parameters and blood viscosity of the three groups are shown in Table 5. There was a proportional correlation between the viscosity and leukocytes and neutrophils in dogs treated with doxycycline.

## 4. Discussion

*E. canis* is a common infection that affects dogs and can show varying clinical signs. However, it is challenging to determine the stage of the disease in naturally infected dogs compared to experimentally infected dogs. In this study, dogs exhibited clinical signs such as fever, lethargy, pale gums, loss of appetite, bleeding, and swollen lymph nodes, characteristic of CME. Other authors have reported similar clinical findings [31]. Clinical, laboratory, serological, and molecular findings are necessary to diagnose canine ehrlichiosis. Early diagnosis is crucial for effective treatment of the disease [32].

The disease can cause significant changes in a dog’s blood count [12,13,33]. In this study, dogs infected with *E. canis* showed alterations in blood cells, particularly in lymphocytes, eosinophils, and platelets. This infection also affected the rheological behavior and cytokine levels of the blood. Although doxycycline treatment improved hematological parameters, no significant changes were observed in blood viscosity levels after the treatment in infected animals.

CME is challenging to diagnose due to its various stages and clinical manifestations. Hematological abnormalities have been reported in infected dogs [34]. Here, the evaluation of leukocytes in infected dogs shows lymphopenia, and no effects on these cells were observed after the doxycycline treatment. A review update about the treatments used for CME in dogs showed in the conclusions that the first line of treatment with doxycycline could be performed within 3 to 4 weeks [25], as used in the present study. Similarly, another study also cites doxycycline treatment with a duration of 3 to 4 weeks [33]. Thus, the duration of the treatment of 4 weeks demonstrates effectiveness for eliminating the infection of *E. canis* in dogs, and it is not necessary to apply the treatment for up to 6 weeks [35].

Interestingly, the concentration of some cells increased due to doxycycline. Hematological alteration in dogs with CME is also reported in the literature [36]. Tetracyclines are drugs that have immunomodulatory effects in addition to antimicrobial effects. Thus, doxycycline is cited as an antibiotic that causes changes in the counts of subpopulations of lymphocytes in dogs with CME and healthy dogs by inhibiting lymphocyte proliferation or inducing cell death [37]. Doxycycline is an antibiotic used to treat ehrlichiosis [25], but it demonstrates this undesirable effect on lymphocytes.

Studies in literature have reported a decrease in eosinophil levels in dogs infected with *E. canis* [38]. However, this study found that treatment corrected eosinopenia despite the initial reduction in eosinophil levels in infected dogs. Regarding thrombocytopenia in infected dogs with *E. canis*, it is a very common alteration described in CME [38]. The use of doxycycline, in addition to the antimicrobial effect on *E. canis* [39], has a specific effect on platelet proliferation [37]. This description was shown in our study by the increase in platelets in the group of dogs with treatment, an effect also reported in other studies [37,40].

Hematological parameters play a vital role in regulating blood viscosity. Variations in the plasma or cellular components can impact resistance to blood flow in the vascular system and affect tissue perfusion [41]. In addition, blood viscosity is crucial for maintaining the rheological properties of blood [20], and studies have correlated changes in viscosity with infections or metabolic diseases [17,21].

Hemorheology is an important tool for characterizing the rheological behavior of the blood of sick dogs, especially diseases that cause changes in hematological parameters. Thus, blood viscosity is a parameter obtained by the hemorheology technique essential to be analyzed along with hematological parameters and serum cytokines to understand the pathophysiological changes in infectious diseases such as CME. Other articles also observed this mechanism that studied obligate intracellular parasites or bacteria and hemorheology changes caused by the infectious process [14,18,21].

This study showed interesting hematological, circulatory, and immunomodulation changes in infected dogs untreated and treated with doxycycline. There was a decrease in the blood viscosity of dogs infected with *E. canis*. Similarly reported in another study with dogs infected with bacteria from the Anaplasmataceae family, PCR-positive animals had lower blood viscosity than noninfected animals [18]. In addition, CME can lead to blood viscosity and circulatory flow alterations and predispose to complications such as thrombosis, the so-called hyperviscosity syndrome [16]. Few studies have shown data about blood viscosity in infectious diseases in dogs. The blood viscosity of dogs infected with *Leishmania* sp. decreases compared to noninfected ones [14]. The treatment with doxycycline for 28 days reduced the blood viscosity. Research has revealed that dogs with *E. canis* infection display hypercoagulation and indicate that after five weeks of treatment with doxycycline, the fibrinolytic effects can be reversed, potentially restoring blood viscosity in infected animals [33]. Further studies are required to explore the optimal duration of doxycycline treatment and its impact on blood viscosity in dogs with *E. canis* infection.

During the acute phase of CME, there is an increase in the pro-inflammatory cytokine TNF-α [42], attracting TNF-producing cytotoxic T lymphocytes. However, there is a decrease in *Ehrlichia*-specific helper T lymphocytes [43]. This study found that TNF-α levels were lower in dogs infected with *E. canis* than in noninfected dogs, regardless of treatment. This decrease in TNF-α production may be due to the direct effect of the antibiotic doxycycline, which inhibits its production [37,40]. It could also be related to the lymphopenia observed in dogs with CME [44]. As time passes after infection, the serum concentration of TNF-α produced by lymphocytes decreases [42], indicating that dogs with CME may be in the subclinical phase of the disease. To improve the immune response to *E. canis* during treatment, a drug with immunobiological action, such as levamisole [45], could activate phagocytosis and improve the immune response to *E. canis* during treatment. 

Notably, animals infected with *E. canis* showed a rise in Il-1β and monocytes when given doxycycline treatment. IL-1β is a significant player in immune responses and inflammation. This cytokine group includes 11 ligands—7 agonists, 3 receptor antagonists, and 1 anti-inflammatory cytokine—and contributes to the resolution of acute inflammation [46,47]. The increase in these cytokines in dogs infected with *E. canis* might be a valuable indicator for diagnosing the acute stage of CME.

Some studies have shown that in dogs with acute *Ehrlichia* sp. infection, there is no significant difference in serum TNF-α and IL-10 levels between infected and noninfected groups [44]. However, another study found that 18 days after infection, there is a significant increase in TNF-α and IL-10 levels in infected dogs, but this difference disappears after 30 days [44]. Similarly, a study on *Babesia rossi*, another intracellular pathogen, showed a difference in TNF-α and IL-10 cytokine levels between groups of infected and noninfected dogs [48].

This study shows that in dogs treated with doxycycline, there is a correlation between IL-10 and leukocytes. This correlation is not present in untreated animals, suggesting that doxycycline has an immunomodulatory effect. IL-10 is an important cytokine that helps balance the Th1/Th2 cell response in dogs with CME. Previous research has shown immunomodulation between TNF-α and IL-10 [44]. In dogs infected with *E. canis*, IL-10 is inverse correlated with blood viscosity, which becomes proportional with doxycycline treatment. In untreated infected dogs, IL-12 also has an inverse correlation with blood viscosity. IL-10 plays a vital role in regulating macrophages and monocytes and suppressing inflammatory cytokines such as IL-1β, IL-6, IL-12, and TNF-α [49]. Another study that correlated viscosity and cytokines demonstrates that Interferon Gamma (IFN-y) correlates with blood viscosity in dogs infected with bacteria from the Anaplasmataceae family, indicating that the higher the level of IFN-y, the greater the value of blood viscosity [18]. Still, this study observed data from infected individuals and not post-treatment with doxycycline.

Data on changes in blood viscosity in dogs with infectious diseases are scarce [14,18]. Blood viscosity in this study correlated with TNF-α values in the uninfected dogs with high serum concentrations of this cytokine. Perhaps high levels play an important role in modulating blood viscosity in dogs. A human study demonstrated that patients with sickle cell disease had higher levels of TNF-α and blood viscosity when compared to healthy individuals [50].

Another hematological parameter that demonstrated an effect on viscosity was the leukocytes and neutrophils in the group treated with doxycycline. White blood cell and platelet counts are reported as components affecting blood viscosity [51]. As the infected group had a lower leukocyte count than the noninfected group, this could explain one of the causes of the decrease in blood viscosity and the decrease in TNF-α levels. Blood components such as erythrocytes, hematocrit, and platelets can influence the variation in blood viscosity [51]. However, no correlations were observed between these parameters and blood viscosity in any of the groups in this study.

To ensure that ehrlichiosis is diagnosed accurately, it is recommended to use multiple diagnostic assays, including serology-based tests like ELISA and molecular assays. Studies have shown that confirming the diagnosis through a combination of serology and PCR testing is effective for canine ehrlichiosis. Real-time PCR can measure bacterial loads and identify specific gene fragments, with sequencing revealing the *Ehrlichia* species responsible for the infection [1,52,53]. However, clinical and hematological evaluations must also be considered. In this study, dogs showing symptoms of CME were initially screened with rapid serological tests and then examined using PCR to detect *E. canis* DNA. However, it is important to note that this approach may have some limitations. Furthermore, the timing of the dogs’ infection was not precisely determined because they were naturally infected animals, which could have influenced some of the laboratory results. More research is needed on alternative diagnostic tests to improve the detection of ehrlichiosis and other markers that may help identify disease progression in naturally infected dogs.

## 5. Conclusions

The findings of this study indicate that dogs with *E. canis* had lower total leukocytes, lymphocytes, eosinophils, platelets, blood viscosity, and TNF-α. The treatment with doxycycline was able to increase platelet and eosinophil levels. However, the drug increased IL-1-β and monocyte levels, which could exacerbate inflammation. Thus, when treating a dog with monocytic ehrlichiosis using doxycycline, it is important to evaluate viscosity parameters and cytokine levels.

## Figures and Tables

**Figure 1 biology-12-01137-f001:**
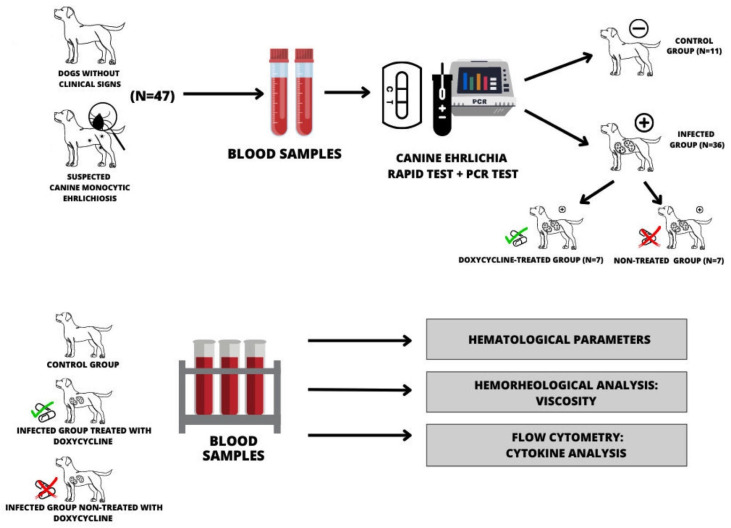
Representative scheme for obtaining samples and experimental design.

**Figure 2 biology-12-01137-f002:**
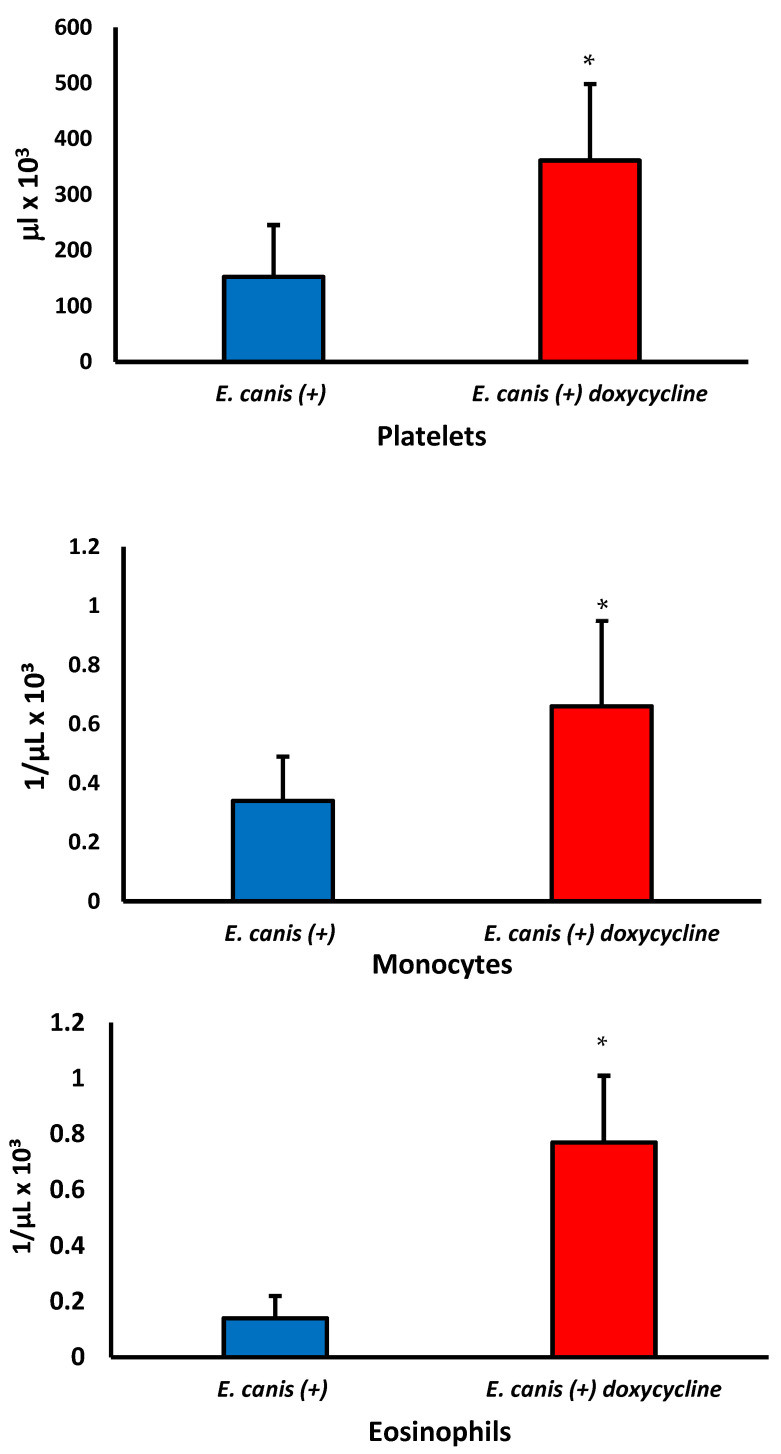
Effect of doxycycline of platelets, monocytes, and eosinophils in blood of dogs with CME untreated (blue color) and treated (red color) treatment. Student *t*-test for two related samples. * differences between before and after treatment. Platelets *p* = 0.0377, monocytes *p* = 0.0303, and eosinophils *p* = 0.0025.

**Figure 3 biology-12-01137-f003:**
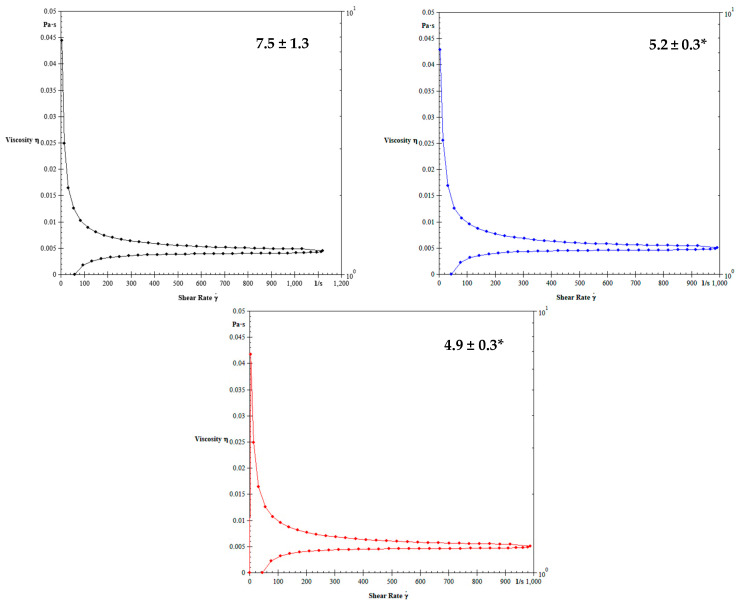
Blood viscosity of the noninfected dogs (black color), infected dogs (blue color), and infected dogs treated with doxycycline (red color). * *p* < 0.05 difference between noninfected dogs (control) with the infected dogs (treated or not with doxycycline). ANOVA test followed by Tukey test was used post-test.

**Figure 4 biology-12-01137-f004:**
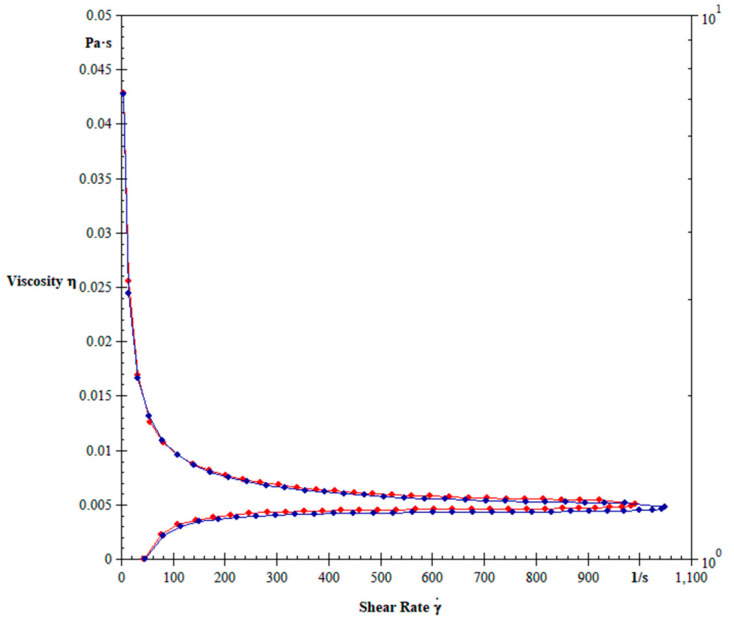
Effect of doxycycline on blood viscosity of dogs with untreated CME (blue color) and treated (red color) with doxycycline. difference between untreated dogs and dogs treated with doxycycline. Student *t*-test for two related samples. *p* < 0.0001.

**Figure 5 biology-12-01137-f005:**
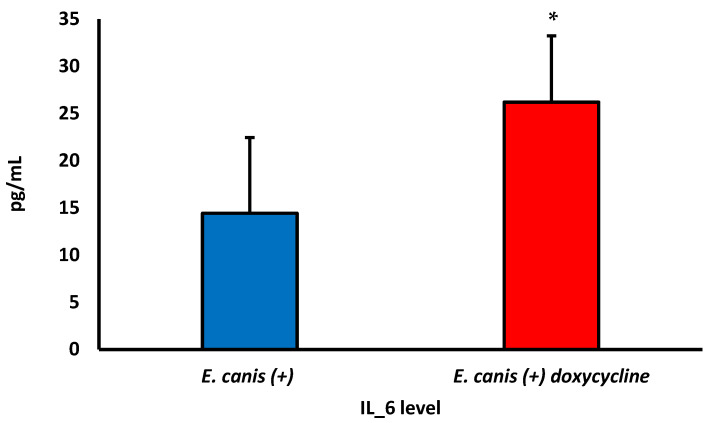
Effect of doxycycline on IL-6 levels of dogs with CME without (blue color) and with (red color) treatment with doxycycline. *p* < 0.0073; * difference between infected and untreated dogs and dogs treated with doxycycline. Student t-test for two related samples.

**Table 1 biology-12-01137-t001:** Hematological values (mean and standard error) in noninfected dogs and infected dogs with *E. canis* untreated and treated with doxycycline.

Parameters	*E. canis* (-)	*E. canis* (+)	*E. canis* (+) Doxycycline	*p*
Erythrocytes (10^6^/μL)	6.42 ± 1.82	5.73 ± 2.08	5.69 ± 1.64	0.9545
Hemoglobin (g/dL)	14.72 ± 3.98	13.23 ± 5.36	14.06 ± 3.86	0.8443
Hematocrit (%)	43.36 ± 11.85	37.58 ± 14.29	38.73 ± 11.06	0.8654
Leukocytes (10^3^/μL)	10.27 ± 2.98 ^a^	8.36 ± 2.49 ^b^	8.70 ± 3.81 ^b^	0.0119
Neuthrophils (10^3^/μL)	6.32 ± 2.21	6.88 ± 2.61	6.35 ± 2.98	0.5564
Lymphocytes (10^3^/μL)	2.84 ± 1.98 ^a^	1.25 ± 0.79 ^b^	1.31 ± 0.77 ^b^	0.0086
Monocytes (10^3^/μL)	0.33 ± 0.12 ^a^	0.30 ± 0.21 ^a^	0.64 ± 0.36 ^b^	0.0257
Eosinophils (10^3^/μL)	0.67 ± 0.78 ^a^	0.26 ± 0.28 ^b^	0.60 ± 0.36 ^a^	0.0290
Platelets (10^3^μL)	211.90 ± 161.28 ^a^	183.68 ± 113.72 ^b^	298.26 ± 178.50 ^c^	0.0124

*E. canis* (-): dogs negative for *Ehrlichia canis*; *E. canis* (+): dogs positive for *Ehrlichia canis*; *E. canis* (+) doxycycline: dogs treated with doxycycline. Averages followed by different letters differ significantly by *p* < 0.05, considering the same parameters. ANOVA test followed by the Tukey was used post-test.

**Table 2 biology-12-01137-t002:** Serum cytokine values (mean and standard error) in noninfected dogs and infected dogs with *E. canis* untreated and treated with doxycycline.

Cytokines (pg/mL)	*E. canis* (-)	*E. canis* (+)	*E. canis* (+) Doxycycline	*p*
IL-1β	25.59 ± 8.61	56.72 ± 24.75 a	55.38 ± 30.76 a	0.0150
IL-6	28.66 ± 14.04	14.42 ± 8.03	23.63 ± 13.85	0.1588
IL-8	20.40 ± 3.60	31.17 ± 20.51	33.95 ± 22.62	0.5416
IL-10	34.39 ± 12.60	30.46 ± 30.59	46.83 ± 26.45	0.3815
IL-12	28.38 ± 9.16	32.47 ± 11.97	25.24 ± 8.86	0.4985
TNF-α	589.75 ± 197.56 a	32.97 ± 37.09 b	340.40 ± 26.65 c	0.0001

*E. canis* (-): dogs negative for *Ehrlichia canis*; *E. canis* (+): dogs positive for *Ehrlichia canis*; *E. canis* (+) doxycycline: dogs treated with doxycycline. Averages followed by different letters differ significantly by *p* < 0.05, considering the same parameters. ANOVA test followed by the Tukey test was used post-test.

**Table 3 biology-12-01137-t003:** Pearson’s linear correlations and values of linear coefficients (r) and statistical significance (*p*) between hematological parameters and serum cytokines in noninfected dogs and dogs infected with *E. canis* either untreated or treated with doxycycline.

Parameters	Group	IL-1β		IL-6		IL-8		IL-10		IL-12		TNF-α	
	*E. canis*	r	*p*	r	*p*	r	*p*	r	*p*	r	*p*	r	*p*
Erythrocytes	negative	−0.024	0.958	−0.270	0.420	−0.619	0.138	−0.457	0.157	0.134	0.773	−0.118	0.728
	positive	−0.248	0.591	−0.190	0.683	−0.711	0.073	−0.244	0.597	−0.738	0.057	−0.660	0.106
	positive + doxycycline	−0.324	0.477	0.156	0.738	−0.166	0.722	0.681	0.092	0.101	0.828	0.142	0.760
Hematocrit	negative	0.073	0.875	−0.528	0.094	−0.579	0.172	−0.466	0.148	0.121	0.795	−0.122	0.719
	positive	−0.360	0.427	−0.166	0.720	−0.745	0.054	−0.316	0.488	−0.783	0.037 *	−0.726	0.064
	positive + doxycycline	−0.578	0.173	−0.165	0.722	0.057	0.902	0.668	0.100	0.219	0.637	−0.113	0.808
Hemoglobin	negative	0.063	0.892	−0.489	0.126	−0.651	0.112	−0.464	0.150	0.057	0.903	−0.062	0.855
	positive	−0.233	0.613	−0.061	0.896	−0.559	0.191	−0.121	0.795	−0.623	0.131	−0.654	0.110
	positive + doxycycline	−0.519	0.231	−0.172	0.711	−0.019	0.967	0.678	0.093	0.239	0.604	−0.111	0.812
Leukocytes	negative	0.113	0.809	0.037	0.912	−0.594	0.159	0.015	0.964	0.353	0.436	−0.463	0.150
	positive	0.215	0.642	0.472	0.284	−0.278	0.545	0.506	0.298	0.148	0.751	−0.095	0.838
	positive + doxycycline	0.500	0.252	0.243	0.599	−0.145	0.755	−0.757	0.048 *	0.318	0.487	0.486	0.268
Neutrophils	negative	0.080	0.863	−0.152	0.654	−0.162	0.727	0.224	0.506	0.282	0.539	−0.396	0.227
	positive	0.119	0.799	0.314	0.492	−0.386	0.391	0.410	0.360	0.068	0.884	−0.183	0.694
	positive + doxycycline	0.686	0.088	0.366	0.418	−0.229	0.619	−0.747	0.053	0.155	0.739	0.525	0.225
Monocytes	negative	0.051	0.913	0.168	0.620	−0.579	0.172	0.296	0.376	−0.001	0.997	−0.226	0.502
	positive	−0.234	0.613	0.022	0.961	−0.075	0.871	−0.466	0.291	−0.273	0.552	−0.004	0.992
	positive + doxycycline	−0.753	0.050	−0.337	0.459	0.191	0.681	0.2749	0.550	0.185	0.691	−0.080	0.863
Lymphocytes	negative	−0.555	0.195	0.190	0.573	0.461	0.297	0.015	0.964	0.512	0.239	−0.073	0.831
	positive	−0.021	0.963	−0.257	0.577	0.187	0.686	−0.399	0.374	−0.233	0.614	0.123	0.792
	positive + doxycycline	−0.313	0.493	−0.368	0.416	−0.077	0.868	−0.201	0.664	0.692	0.084	0.095	0.838
Eosinophils	negative	0.631	0.128	0.065	0.849	−0.297	0.517	0.144	0.671	−0.471	0.285	−0.343	0.300
	positive	0.506	0.246	0.388	0.389	0.177	0.703	0.523	0.227	0.784	0.036 *	0.847	0.016 *
	positive + doxycycline	0.429	0.335	−0.029	0.950	−0.111	0.811	−0.001	0.997	−0.560	0.190	0.077	0.868
Platelets	negative	−0.184	0.692	−0.040	0.906	−0.388	0.388	0.240	0.645	−0.090	0.846	−0.035	0.918
	positive	−0.179	0.699	−0.338	0.457	−0.013	0.977	−0.014	0.976	−0.293	0.523	−0.597	0.156
	positive + doxycycline	0.160	0.730	−0.175	0.706	0.409	0.361	0.113	0.808	−0.005	0.991	−0.600	0.154

(negative) dogs negative for *Ehrlichia canis*; (positive) dogs positive for *Ehrlichia canis*; positive + doxycycline: dogs positive for *Ehrlichia canis* and treated with doxycycline. (*): statistical significance.

**Table 4 biology-12-01137-t004:** Pearson’s linear correlations and values of linear coefficients (r) and statistical significance (*p*) between serum cytokines and blood viscosity parameters in noninfected dogs and dogs infected with *E. canis* untreated or treated with doxycycline.

Cytokine	Group	Blood Viscosity	
	*E. canis*	r	*p*
IL-1β	negative	−0.195	0.673
	positive	−0.762	0.046 *****
	positive + doxycycline	0.208	0.654
IL-6	negative	0.134	0.694
	positive	−0.082	0.860
	positive + doxycycline	0.044	0.924
IL-8	negative	−0.395	0.379
	positive	−0.697	0.081
	positive + doxycycline	0.153	0.742
IL-10	negative	0.294	0.379
	positive	−0.797	0.031 *****
	positive + doxycycline	0.882	0.019 *****
IL-12	negative	0.135	0.772
	positive	−0.838	0.018 *****
	positive + doxycycline	0.598	0.155
TNF-α	negative	−0.930	>0.001 *****
	positive	−0.672	0.097
	positive + doxycycline	0.142	0.760

(negative) dogs negative for *Ehrlichia canis*; (positive) dogs positive for *Ehrlichia canis*; positive + doxycycline: dogs positive for *Ehrlichia canis* treated with doxycycline. (*): statistical significance.

**Table 5 biology-12-01137-t005:** Pearson’s linear correlations and values of linear coefficients (r) and statistical significance (*p*) between hematological and blood viscosity parameters in noninfected dogs and dogs infected with *E. canis* untreated or treated with doxycycline.

Hematological Parameters	Group	Blood Viscosity	
	*E. canis*	r	*p*
Erythrocytes	negative	0.022	0.946
	positive	0.647	0.115
	positive + doxycycline	−0.266	0.563
Hematocrit	negative	0.026	0.939
	positive	0.722	0.066
	positive + doxycycline	−0.316	0.489
Hemoglobin	negative	−0.033	0.922
	positive	0.558	0.192
	positive + doxycycline	−0.350	0.441
Leukocytes	negative	0.378	0.2504
	positive	−0.274	0.552
	positive + doxycycline	0.917	0.003
Neutrophils	negative	0.351	0.289
	positive	−0.272	0.554
	positive + doxycycline	0.804	0.029 **-**
Monocytes	negative	0.269	0.423
	positive	0.577	0.174
	positive + doxycycline	−0.036	0.938
Lymphocytes	negative	0.032	0.924
	positive	0.439	0.324
	positive + doxycycline	0.523	0.228
Eosinophils	negative	0.247	0.462
	positive	−0.541	0.209
	positive + doxycycline	−0.578	0.173
Platelets	negative	0.087	0.797
	positive	−0.180	0.699
	positive + doxycycline	−0.332	0.465

(negative) dogs negative for *Ehrlichia canis*; (positive) dogs positive for *Ehrlichia canis*; positive + doxycycline: dogs positive for *Ehrlichia canis* treated with doxycycline.

## Data Availability

If requested, the authors will provide the data supporting this study’s interpretations.

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
