# Peer review of "Effects of Doxycycline Treatment on Hematological Parameters, Viscosity, and Cytokines in Canine Monocytic Ehrlichiosis"

_biology, 2023, doi:10.3390/biology12081137_

Round 1

Reviewer 1 Report

This manuscript analyzed the hematological parameters, blood viscosity and cytokines of dogs infected by Ehrlichia canis with doxycycline treatment, found some correlations between these results, and suggested to evaluate viscosity and cytokine levels when treating dogs. It is a straightforward study and the manuscript is basically clear.

Some points to correct or improve:

1.      Some typos/formats: line 43 “Riphicephalus”-> Rhipicephalus; line 102 “eosinophils” -> ”monocytes”; line 170 “untread” -> untreated; line 364 “pb” -> bp; line 371 “Sybr” -> SYBR; line 386 “Hemorrheological” -> Hemorheological; use either Leucocytes or “Leukocytes”, rather than both in the manuscript.

2.      Line 92: “when the dogs were treated with doxycycline” compared to which group?

3.      Line 100 and 150: no difference should be “no significant difference”.

4.      Table 1 and 2: what do these superscript “a” “b” “c” “*” stand for? I don’t understand the unit “1/μL x 10³”, which may be “x 10³/μL”? There is no unit at all in Table 2? There are 3 groups in each row, so which 2 groups are all these P values calculated between? In addition, please remove “P=” and just leave the numbers.

5.      Figure 1 and 4: are these data from Table 1 and 2? Look not quite consistent with the data in the tables.

6.      Figure 3: are these 2 curves the same as in Figure 2 and just overlay? If so, it can be put as a panel in Figure 2. These 2 curves look very similar, so I don’t understand why the different is so significant (P<0.0001).

7.      I think “untreated and treated” or “with and without treatment” is a more accurate (de facto) description than “before and after treatment”.

8.      Table 3: “R” should be “r”? 3 (not 4) decimal places are good enough for all numbers and will make the table cleaner and easier to read (also for Table 4 and 5).

9.      Line 168: in the Table 3, it shows “r = -0.783”, which means actually “inversely proportional correlation”?

10.  Line 307: the gene is 0503 or 0701?

11.  Line 262 and 343: please explain the reason to choose the time point of 28 days, since other study used 5 weeks.

Author Response

Reviewer 1

Thank you very much for the suggestions for improving the manuscript. All changes have been made and are described below.

Some points to correct or improve:

  1. Some typos/formats: line 43 “Riphicephalus”-> Rhipicephalus; line 102 “eosinophils” -> ”monocytes”; line 170 “untread” -> untreated; line 364 “pb” -> bp; line 371 “Sybr” -> SYBR; line 386 “Hemorrheological” -> Hemorheological; use either “Leucocytes” or “Leukocytes," rather than both in the manuscript.

R: Was adjusted according to suggestion.

  1. Line 92: “When the dogs were treated with doxycycline” compared to which group?

R: Infected animals were compared before and after treatment. We improved the phrase.

  1. Lines 100 and 150: “No difference” should be “no significant difference

."

R: It was modified

  1. Table 1 and 2: what do these superscript “a” “b” “c” “*” stand for? I don’t understand the unit “1/μL x 10³" which may be “x 10³/μL”? There is no unit at all in Table 2. There are 3 groups in each row, so which 2 groups are all these P values calculated between? In addition, please remove “P=” and just leave the numbers.

R: Subscripts with different letters indicate differences between groups. The units have been modified. The unit (pg/mL) was placed on the table. Comparisons were made using the ANOVA test comprising three or more groups, so the P value refers to comparisons between the three groups. The P has been removed from the table. More information was added to the tables.

  1. Figure 1 and 4: are these data from Table 1 and 2? Look not quite consistent with the data in the tables.

R: Figures 1 and 4 compare infected animals before and after treatment using the t-test for two related samples to assess the effects of treatment. More information was added to the figures.

  1. Figure 3: are these 2 curves the same as in Figure 2 and just overlay? If so, it can be put as a panel in Figure 2. These 2 curves look very similar, so I don’t understand why the difference is so significant (P<0.0001).

R: Figure 2 shows comparisons were made using the ANOVA test compared to the three groups. Figure 3 compares infected animals before and after treatment using the t-test for two related samples to assess the effects of treatment. More information was added in the figure.

        The significant difference can be seen at the end of the curve, where the animal's blood before treatment (X-axis scale 1000 times usually in rheological analysis) is more viscous than after treatment.

  1. I think “untreated and treated” or “with and without treatment” is a more accurate (de facto) description than “before and after treatment."

R: was modified according to suggested.

  1. Table 3: “R” should be “r”? 3 (not 4) decimal places are good enough for all numbers and will make the table cleaner and easier to read (also for Tables 4 and 5).

R; The tables were adjusted according to suggestions.

  1. Line 168: in Table 3, it shows “r = -0.783”, which means actually “inversely proportional correlation”?

R: It was modified

  1. Line 307: the gene is 0503 or 0701?

R: the gene is 0503 and was corrected.

  1. Line 262 and 343: please explain the reason for choosing the time point of 28 days since the other studies used 5 weeks.

        A review update about the treatments used for CME in dogs showed in the conclusions that the first line of treatment with doxycycline could be done within 3 to 4 weeks duration, as used in the present study. Similarly, another study also cites doxycycline treatment with 3 to 4 weeks duration. Thus, the duration of the treatment of 4 weeks demonstrates effectiveness for eliminating the infection by E. canis in dogs, not being necessary to apply the treatment for up to 6 weeks [35]. This information was added to the discussion.

Mylonakis, M.E.; Harrus, S.; Breitschwerdt, E.B. An update on the treatment of canine monocytic ehrlichiosis (Ehrlichia canis). Vet J 2019, 246, 45-53.

Shropshire, S.; Olver, C.; Lappin, M. Characteristics of hemostasis during experimental Ehrlichia canis infection. J Vet Intern Med 2018, 32, 1334-1342.Harrus S.; Waner, T.; Aizenberg, I.; Bark, H. Therapeutic effect of doxycycline in experimental subclinical canine monocytic ehrlichiosis: evaluation of a 6-week course. J Clin Microbiol. 1998, 36(7), 2140-2142.

Reviewer 2 Report

General comments: The manuscript tiled “Effects of doxycycline treatment on hematological parameters, viscosity and cytokines in canine monocytic ehrlichiosis” explores a seldom focused on subject. The article is mostly well written but could benefit from language revision. The method used are sound as far as I understood them. I have no major concerns regarding the manuscript, but several smaller issues that I think need to be addressed.

Specific comments:

Line 29: Should likely be infected with Ehrlichia instead of infected with IL-10 and IL-12.

Line 30: Eosinophil count correlated, not eosinophils, correct?

Line 52: There should be a comma after uremia.

Line 52-53: Subclinical disease, by definition, does not show clinical signs of disease. Hence this is a bit redundant.

Line 54: What does tutors refer to? Do you mean owners?

Line 84-88: Since the number of dogs was quite low, I would urge the authors to add the exact number of patient (e.g. 2% out of 36 is 0.7 dogs, while 3% out of 36 is 1.08 dogs).

Table 4 and 5: please add a caption for what the asterisk (*) notes.

Line 209: The c in Canis should not be capitalized.

Line 223: Please rephrase or consider removing. Many diseases can cause hematological abnormalities, but the “Since” in this sentence makes it sound like it is somehow exclusive to these diseases.

Line 225-226: Please rephrase. While the concentration of some cells increased due to doxycycline, for other this was likely not due to doxycycline directly.

Line 253-254: What decreases in dog infected with EC?

Line 255: Patients are usually not PCR-reactive, rather PCR-positive.

Line 333: Was selection truly random (using e.g. a random number generator) or was it haphazard? Please rephrase accordingly.

Line 335: Inclusion criteria should be more specific. Clinical signs of CME are numerous. How many symptoms were needed and how severe were they to be classified as signs of CME? You could use the sentence on lines 211-213 if this is true for this study.

Line 348: N-7 should be N=7, correct?

Line 415-417: While the conclusion is generally sound, I do not see how the authors justify the importance of evaluating viscosity parameters and cytokine levels in dogs with MCE treated with doxycycline. Considering that not treating a dog with CME is rarely an option and tests measuring viscosity parameters and cytokine levels are very rare, is this a feasible requirement for practitioners? Since no interventions were done or even suggested to increase/decrease either, why is it important to monitor?

Some language editing is still needed, although generally well written and comprehensible.

Author Response

Reviewer 2

Thank you very much for the suggestions for improving the manuscript. All changes have been made and are described below.

General comments: The manuscript titled "Effects of doxycycline treatment on hematological parameters, viscosity and cytokines in canine monocytic ehrlichiosis" explores a seldom focused subject. The article is mostly well-written but could benefit from language revision. The method used is sound as far as I understood them. I have no major concerns regarding the manuscript, but several smaller issues that I think need to be addressed.

Specific comments:

Line 29: Should likely be infected with Ehrlichia instead of infected with IL-10 and IL-12.

R: It was modified

Line 30: Eosinophil count correlated, not eosinophils, correct?

R: It was corrected

Line 52: There should be a comma after uremia.

R: OK.

Line 52-53: Subclinical disease, by definition, does not show clinical signs of disease. Hence this is a bit redundant.

R: In the subclinical phase, clinical signs may often not be observed or not detected during the clinical evaluation due to the apparent physical health of the dog, but splenomegaly and intermittent fever may occur, or even signs not commonly reported by the dog’s owners. This information was added to the manuscript.

  1. Ramakant, R.K.; Verma, H.C.; Diwakar, R.P. Canine ehrlichiosis: A review. J Entomol Zool 2020, 8, 1849-1852.
  2. Mylonakis, M.E.; Theodorou, K.N. Canine monocytic ehrlichiosis: an update on diagnosis and treatment. Act Vet 2017, 67, 299-317.

Line 54: What do tutors refer to? Do you mean owners?

R: It was corrected

Line 84-88: Since the number of dogs was quite low, I would urge the authors to add the exact number of patients (e.g., 2% out of 36 is 0.7 dogs, while 3% out of 36 is 1.08 dogs).

R: Was adjusted according to suggested.

Tables 4 and 5: please add a caption for what the asterisk (*) notes.

R: Was added.

Line 209: The c in Canis should not be capitalized.

R: was adjusted

Line 223: Please rephrase or consider removing. Many diseases can cause hematological abnormalities, but the “Since” in this sentence makes it sound like it is somehow exclusive to these diseases.

R: The phrase was modified

Line 225-226: Please rephrase. While the concentration of some cells increased due to doxycycline, for others, this was likely not due to doxycycline directly.

R: Was adjusted

Line 253-254: What decreases in dog infected with EC?

R: The Blood viscosity was added in the text.

Line 255: Patients are usually not PCR-reactive, rather PCR-positive.

R: It was modified.

Line 333: Was selection truly random (using e.g., a random number generator), or was it haphazard? Please rephrase accordingly.

R: The selection was haphazard and was modified in the text

Line 335: Inclusion criteria should be more specific. Clinical signs of CME are numerous. How many symptoms were needed, and how severe were they to be classified as signs of CME? You could use the sentence on lines 211-213 if this is true for this study.

R: Dogs with at least three clinical signs were submitted to serological tests and confirmed by PCR. This criterion was further defined in the materials and methods section.

Line 348: N-7 should be N=7, correct?

R: it was corrected.

Line 415-417: While the conclusion is generally sound, I do not see how the authors justify the importance of evaluating viscosity parameters and cytokine levels in dogs with MCE treated with doxycycline. Considering that not treating a dog with CME is rarely an option and tests measuring viscosity parameters and cytokine levels are very rare, is this a feasible requirement for practitioners? Since no interventions were done or even suggested to increase/decrease either, why is it important to monitor?

R: The importance of monitoring and association with viscosity and cytokines in dogs with ECM treated with doxycycline is justified because these parameters can help diagnose and monitor the evolution of treated and untreated animals. It can also serve as a guide to treatment options or new therapeutic approaches.

Reviewer 3 Report

The authors present a study investigating the hematological parameters, blood viscosity and cytokines of dogs infected by Ehrlichia canis before (n=7) and after treatment with doxycycline (n=7) and in healthy dogs (n=11).

This study is pertinent and novel because there is a scarce information on data about blood viscosity in infectious diseases in dogs. Congratulations to the authors for this work.

The manuscript is generally well written and presented. I suggest the following modifications and considerations:

Abstract

Line 19. You should write the total number of dogs finally included in the study as they were 25: 11 healthy dogs and 14 E. canis positive dogs.

Lines 20-22. Please, add the number of dogs included in the study per group as follow: “including a control group of healthy dogs (n=11), a group of infected dogs without treatment (n=7), and a group of infected dogs treated with doxycycline (n=7)”

Line 26. Please modify the following sentence: “The eosinophils and platelets decreased in dogs with Ehrlichia canis infections” as “The eosinophils and platelets decreased in dogs with Ehrlichia canis infection without treatment.”

Keywords: Doxycycline should be added to the Keywords. Maybe instead of blood?

Introduction

Line 44. Please rephrase the following sentence: “from ticks such as Riphicephalus sanguineus” as “from hard ticks (Ixodidae) such as Riphicephalus sanguineus”.

Results

Line 91. Please modify the following sentence: “The eosinophils and platelets decreased in dogs with Ehrlichia canis infections” as “The eosinophils and platelets decreased in dogs with Ehrlichia canis infection without treatment.”

Table 1. Please, add the number of dogs included per group. E.g. E. canis (-) n=11.

And explain in the table caption the meaning of the letters a, b and c superscript. And the same for table 2.

Line 131. Review the sentence: “The mean blood viscosity of animals infected by E. canis, independently of treatment, decreased blood viscosity”. I think that it would be as “The mean blood viscosity of animals infected by E. canis decreased, independently of treatment.”

Lines 167-168. Review the sentence: “IL-12 showed a 167 directly proportional correlation with hematocrit in infected and untreated dogs” According to data provide in table 3 it would be an inversely proportional correlation (r= -0.783).

Discussion

Line 209. Please change E. Canis as E. canis.

Lines 219-220. “Although doxycycline treatment improved hematological parameters, no significant effects were observed on the blood viscosity and cytokines in infected animals”

Do you mean after the treatment in infected animals? Please, clarify it.

Line 234. “There was no correction of eosinopenia due to the effect of the treatment”. But eosinopenia was significantly higher in the treated infected dogs? (Figure 1 and lines 102-103). Please, clarify it.

Line 253. “There was a decrease in 253 dogs infected with E. cani” I think there is something missing in this sentence…

Lines 312-313. “As the infected groups had a lower leukocyte count than the non-infected groups.”

Please replace “infected groups” as “infected group” and “noninfected groups” as “noninfected group.”

Methodology

Line 342. Please include the number of animals that were not included in the study of the 36 E. canis positive dogs. In addition, how many dog were finally treated in total? Could you also add this information?

Line 343. “10mg/kg every 12 hours for 28 consecutive days” In the abstract section appears “10mg/kg dose for 28 consecutive days”. Please, clarify it.

Line 344. When did the infected treated dogs return to do the blood sample collection? Was it once the treatment was completed (on day 28 after starting the treatment)? Were there different time points depending on the dog?

Please add this information.

Line 348. Were all the treated dogs in the acute phase of the disease?

And for the animals finally included in the study. Were other vector-borne disease analysed? Because co-infection in infected animals as well as subclinical infections for healthy dogs could affect the haematological and inflammatory parameters measured. Please clarify it in the manuscript.

Line 407. Did you run normality test before t-student test?

General comment

This study provides interesting data, and it is very valuable that it was carried out with naturally infected dogs. It contributes to reduce the number of experimental animals used in research studies.

However, one of the main limitations is that the time when the dogs became infected is unknown (as this study evaluates naturally infected animals). Therefore, some of the laboratory findings could be different in E. canis-infected dogs depending on which course of infection they were on. I think you should include this limitation in the discussion.

Author Response

Reviewer 3

Thank you very much for the suggestions for improving the manuscript. All changes have been made and are described below.

The authors present a study investigating the hematological parameters, blood viscosity, and cytokines of dogs infected by Ehrlichia canis before (n=7) and after treatment with doxycycline (n=7) and in healthy dogs (n=11). 

This study is pertinent and novel because there is scarce information on data about blood viscosity in infectious diseases in dogs. Congratulations to the authors for this work. 

The manuscript is generally well-written and presented. I suggest the following modifications and considerations: 

Abstract 

Line 19. You should write the total number of dogs finally included in the study as they were 25: 11 healthy dogs and 14 E. canis positive dogs. 

R: The final number of dogs in this study was added to um abstract.

Lines 20-22. Please, add the number of dogs included in the study per group as follow: “including a control group of healthy dogs (n=11), a group of infected dogs without treatment (n=7), and a group of infected dogs treated with doxycycline (n=7)." 

The number was added.

Line 26. Please modify the following sentence: “The eosinophils and platelets decreased in dogs with Ehrlichia canis infections” as “The eosinophils and platelets decreased in dogs with Ehrlichia canis infection without treatment.” 

R: it was modified.

Keywords: Doxycycline should be added to the Keywords. Maybe instead of blood? 

R: was changed

Introduction 

Line 44. Please rephrase the following sentence: “from ticks such as Riphicephalus sanguineus” as “from hard ticks (Ixodidae) such as Riphicephalus sanguineus." 

R: the phrase was adjusted.

Results 

Line 91. Please modify the following sentence: “The eosinophils and platelets decreased in dogs with Ehrlichia canis infections” as “The eosinophils and platelets decreased in dogs with Ehrlichia canis infection without treatment.” 

R: the phrase was adjusted.

Table 1. Please, add the number of dogs included per group. E.g. E. canis (-) n=11.

R: The number of dogs was added to the table. 

And explain in the table caption the meaning of the letters a, b and c superscript. And the same for Table 2.

R: Subscripts with different letters indicate differences between groups. This information was added to the figures.

Line 131. Review the sentence: “The mean blood viscosity of animals infected by E. canis, independently of treatment, decreased blood viscosity.” I think that it would be as “The mean blood viscosity of animals infected by E. canis decreased, independently of treatment.”

 R: It was modified.

Lines 167-168. Review the sentence: “IL-12 showed a 167 directly proportional correlation with hematocrit in infected and untreated dogs” According to data provided in Table 3, it would be an inversely proportional correlation (r= -0.783).

R: It was corrected.

Discussion

Line 209. Please change E. Canis as E. canis.

R: It was changed.

Lines 219-220. “Although doxycycline treatment improved hematological parameters, no significant effects were observed on the blood viscosity and cytokines in infected animals.”

 Do you mean after the treatment of infected animals? Please, clarify it.

 R: Yes, after the treatment. The text was modified.

Line 234. “There was no correction of eosinopenia due to the effect of the treatment.” But eosinopenia was significantly higher in the treated infected dogs. (Figure 1 and lines 102-103). Please, clarify it.

R: Many thanks for your observation. It was corrected in the text.

Line 253. “There was a decrease in 253 dogs infected with E. cani” I think there is something missing in this sentence…

 R: The sentence was modified.

Lines 312-313. “As the infected groups had a lower leukocyte count than the noninfected groups.”

Please replace “infected groups” as “infected group” and “noninfected groups” as “noninfected group.”

R: It was corrected.

Methodology

Line 342. Please include the number of animals that were not included in the study of the 36 E. canis-positive dogs. In addition, how many dogs were finally treated in total? Could you also add this information?

 R: This information was added in the material methods section.

Line 343. “10mg/kg every 12 hours for 28 consecutive days” In the abstract section appears “10mg/kg dose for 28 consecutive days”. Please, clarify it.

R: It was added in the abstract the” every 12 hours.”

 .

Line 344. When did the infected treated dogs return to do the blood sample collection? Was it once the treatment was completed (on day 28 after starting the treatment)? Were there different time points depending on the dog?

Please add this information.

 R: The dogs returned for blood collection immediately after completing doxycycline treatment. This information was added to the methodology according to suggest.

Line 348. Were all the treated dogs in the acute phase of the disease?

And for the animals finally included in the study. Were other vector-borne diseases analysed? Because co-infection in infected animals, as well as subclinical infections for healthy dogs,, could affect the hematological and inflammatory parameters measured. Please clarify it in the manuscript.

 R: We included dogs with at least three clinical signs of CME, regardless of their disease stage, as they were naturally infected. Additionally, we conducted tests for Leishmania sp and Babesia on all dogs. This information was clarified in the manuscript. 

Line 407. Did you run normality test before t-student test?

R: A D'Agostino normality test and variance analysis (ANOVA) were used, followed by Tukey’s test between animals noninfected, infected, and infected treated. The student t-test for two related samples was used to compare before and after dogs treated with doxycycline. This information was clarified in the text.

General comment

This study provides interesting data, and it is very valuable that it was carried out with naturally infected dogs. It contributes to reducing the number of experimental animals used in research studies.

However, one of the main limitations is that the time when the dogs became infected is unknown (as this study evaluates naturally infected animals). Therefore, some of the laboratory findings could be different in E. canis-infected dogs depending on which course of infection they were on. I think you should include this limitation in the discussion.

R: The limitation was added in the discussion.